# Prevalence and determinants of late-stage presentation among cervical cancer patients, a systematic review and meta-analysis

**Tiwabwork Tekalign** [ID]*, **Mister Teshome** [ID]

School of Nursing, College of Health Science and Medicine, Wolaita Sodo University, Wolaita, Ethiopia

* tiwabworkt@gmail.com

**Data Availability Statement:** The data underlying the results presented in the study are available from journal and All relevant data are within the manuscript and its Supporting Information files.

## Abstract

### Background

An estimated 570,000 women were diagnosed with cervical cancer worldwide, and about 311,000 women died from the disease. Cervical cancer is possibly the most curable human cancer; if detected at the precancerous stage. Additionally, early diagnosis and management other factors are essential to decrease mortality rate among those patients. So this review was aimed to identify the prevalence and determinants of late-stage presentation among cervical cancer patients.

### Methods

A systematic search had carried out on PubMed, EMBASE, MEDLINE, Cochrane, Scopus, Web of Science CINAHL, and manually on Google Scholar. This meta-analysis follows the Preferred Reporting Items for Systematic Reviews and Meta-Analyses (PRISMA) guidelines. The modified Newcastle-Ottawa Scale (NOS) was used to assess the quality of each study. A meta-analysis was done using a random-effects method using the STATA™ Version 14 software.

### Result

Twenty-five studies from 3 world regions with 53,233 participants were enrolled in this meta-analysis. The overall estimated global pooled prevalence of late-stage presentation among cervical cancer patients with a random-effects model was 60.66% (95% CI: 56.27, 65.06). The subgroup analysis revealed that the prevalence of late-stage presentation was 62.60% in Africa, 69.30% in Asia, 46.51% in Europe, and 50.16% in North America. Educational status (p = 0.031) and place of residence (p = 0.004) are determinants of late-stage presentation.

### Conclusion

The results of this meta-analysis indicated that the prevalence of late-stage presentation of cervical cancer is substantially high. Place of residence and educational status were significantly associated with late-stage presentation. Health care organizations should work on

**Funding:** The authors received no specific funding for this work.

**Competing interests:** The authors have declared that no competing interests exist.

**Abbreviations:** CI, Confidence Interval; NOS, Newcastle Ottawa Scale; OR, Odds Ratio; PRISMA, Preferred Reporting Items for Systematic Reviews and Meta-Analyses; WHO, World Health Organization.

early screening, management, and on increasing community awareness to minimize late stage at presentation among those patients.

## Introduction

World Health Organization (WHO) reports that cervical cancer is the fourth most common cancer in women. An estimated 570, 000 women were diagnosed with cervical cancer worldwide from them around 311, 000 women died from the disease in 2018. Cervical cancer is possibly the foremost curable human cancer, if detected at the precancerous stage [1]. The challenge is that 80% of women in the developing countries seek treatment after they have developed signs and symptoms [2].

Presentation of cervical cancer in a sophisticated stage of disease is the outcome of multiple complex factors including availability of health services for screening, diagnosis, and other cultural and social issues [3]. Reports showed late stage at diagnosis is correlated with lower survival rates in cervical cancer patients [4–6].

Advanced cervical cancer is one among of the main causes of cancer related mortality in women because of poor access to appropriate management especially, in low- and medium income countries. One amongst the foremost important prognostic factors for cervical cancer is how early the disease is detected and how far it's spread. Recently, delay in diagnosis and treatment continues to be the leading obstacle to overcome in the fight to cure cancer [7–9].

Several studies worldwide have investigated the factors related to delayed diagnosis and disparities in its fatality rate in different racial, geographic and socio-economic groups [10–13]; is also knowledge on pooled determinants of delays for this cancer may be useful in establishing comprehensive preventative strategies.

The concept of delayed diagnosis of cervical cancer is categorized in four components including patient delay, health care provider delay, referral delay and system delay. In most countries of the globe, especially in developing countries patients and health care providers delay have more crucial role [14–16].

In Africa, high incidence of cervical cancer has been reported at rates exceeding 50 per 100,000 populations [17]. In sub-Saharan Africa, cervical cancer is the second commonest cause of cancer morbidity and the leading reason for a mortality over 577,000 deaths annually; the same is true in Eastern Africa [18].

Quick scale-up of immunization and double lifetime uterine cervix screening in the world could prevent up to 13.4 million malignancies over the long run half century [19, 20]. Additionally, early diagnosis and managing other factors will decrease mortality and also the prevalence among those patients [21]. So, this systematic review and meta-analysis aimed is to identify pooled prevalence and determinants of late-stage presentation among cervical cancer patients.

## Objective of the review

- To determine pooled prevalence of late stage presentation among cervical cancer patient

- To identify determinants of late stage presentation among cervical cancer patient

## Methods

### Inclusion and exclusion criteria

This systematic review and meta-analysis included all types of studies conducted in different regions of the world which reports the prevalence and determinants of late-stage presentation

among cervical cancer patients, regardless of women's age until July, 17, 2021; whereas, studies which were not fully accessed in which an attempt was made to contact the corresponding author and studies with methodological problems were excluded.

## Information sources, search strategy, and study selection

The studies were retrieved through manual and electronic searches. The databases systematically searched were; PubMed, Scopus, Web of Science, institutional repositories, Academic Search Premier, and manually from reference lists of the previous study. Electronic database searching followed by reference lists search used to identify studies; then exported into End-Note version 7.0 to remove duplicates. Screening of titles and abstracts was done by authors independently. The Cochrane acronym POCC, which stands for population, Condition, and Context, was used to decide on all keywords. The keywords used were, "cervical cancer patient, cervical cancer cases, Late-stage presentation, late diagnosis, delayed diagnosis, advanced disease, early diagnosis, delayed presentation, late tumor stage, prolonged time to diagnosis, delayed care seeking, cancer presentation, delayed access to care, stage of diagnosis, delayed treatment initiation, patient delay, delays in diagnosis, stage at diagnosis, advanced disease at presentation, late-stage cervical cancer, and advanced stage at diagnosis. Finally, this meta-analysis was reported under the Preferred Reporting Items for Systematic Reviews and Meta-analyses (PRISMA) statement 2020 guidelines [22]. We registered the protocol on Prospero (ID: CRD42021284177).

## Data collection process and data items

Data extraction was done by both authors (TT and MT) independently by using a data extraction format prepared in a Microsoft Excel 2013 spreadsheet. The extracted data were: the Author's name, publication year, country, design, sample size, the prevalence of late-stage presentation, and associated factors with their odds ratio.

## The outcome of the review

The primary outcome of this review was the prevalence of late-stage presentation. The second outcome of this review was determinants of late-stage presentation with their odds ratio.

## Quality assessment

The modified Newcastle-Ottawa Scale (NOS) for cross-sectional studies was used to assess the quality of studies [23]. Studies that scored five and more on the NOS were included [24]. Any disagreement while data extractions were resolved through discussion.

## Publication bias and heterogeneity

Funnel plot and Egger's test had used to assess publication bias. A p-value< 0.05 had used to declare the statistical significance of publication bias. $I^2$-statics were computed to assess heterogeneity among reported prevalence $I^2$ test statistics had used to check the heterogeneity of studies. In which if, < 50 declared as low, 50–75% as moderate, and > 75% as having high heterogeneity [25].

## Data synthesis and analysis

A random-effects meta-analysis model was used to estimate the Der Simonian and Laird's pooled effect to show heterogeneity. Subgroup analysis was conducted to adjust random variation between point estimates of original study and investigate how failure fluctuates across

subgroup participants. Outlier within the included articles was checked using sensitivity analysis. Publication bias across studies was assessed using funnel plot and egger's regression test, at $P < 0.05$ to indicate publication bias. Forest plot format was used to present the point prevalence and 95% CIs. In this plot, the weight of study indicated by the size of each box, while each crossed line referred to 95% confidence interval. For the secondary outcomes, odds ratio was used to determine the association between late stage presentation and associated factors. STATA™ V14 software was used to carry out the all Meta-analysis.

## Results

### Study selection

Initially, a total of 64,059 studies had retrieved from the databases and manual searching. From this, 33,547 duplicates were found and removed. The rest, 35,771 articles' were screened by their titles and abstracts. Then 29,835 were irrelevant and removed. Finally, 81 full-text articles were assessed for eligibility, and then due to failure to report the outcome of interest 56, articles were excluded. Finally, a total of 25 studies was fulfilled the inclusion criteria and enrolled in the study. The detailed retrieval process is shown in (Fig 1).

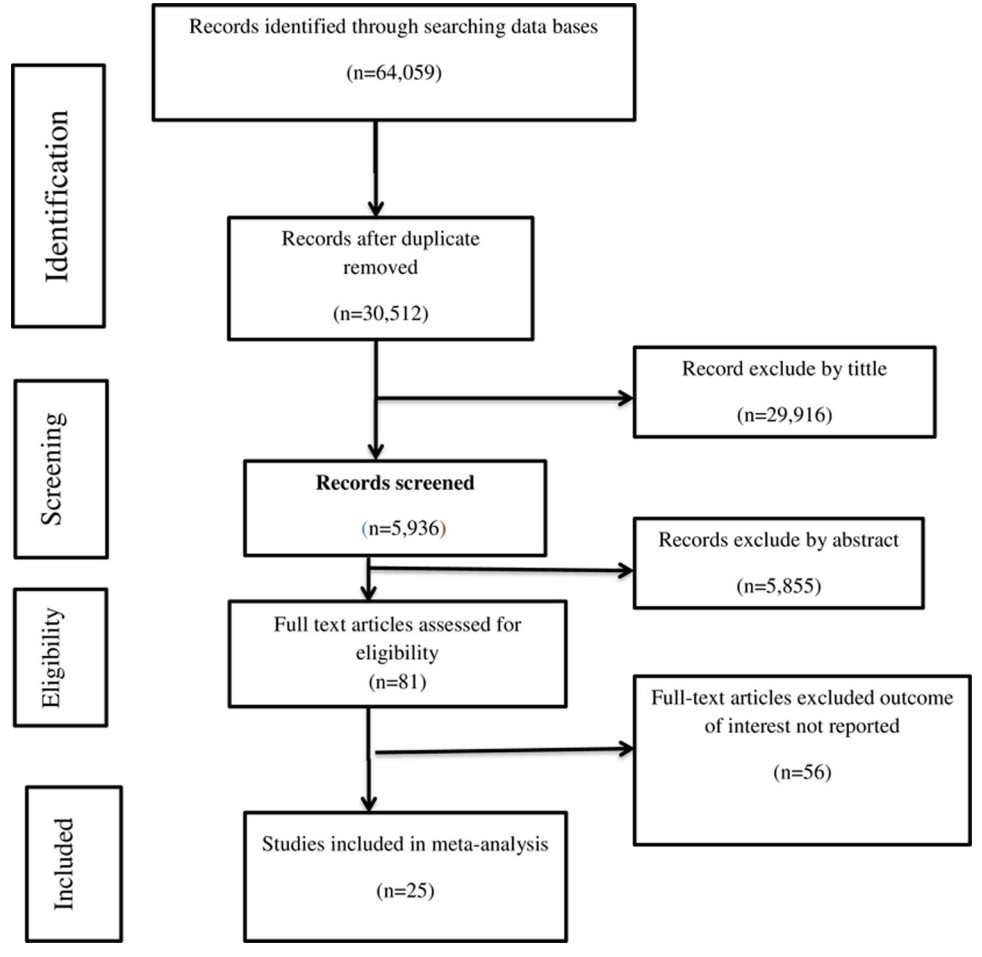

**Fig 1. PRISMA flowchart diagram of the study selection.**

## Study characteristics

The 25 studies [26–50] included 53,233 participants. Most of the included studies were cross-sectional studies and the sample size ranged from 50 [33] to 13624 [50]. Most studies were conducted in Ethiopia. Among the included studies, prevalence of late stage presentation among cervical cancer patients were ranged from 28 [42] to 89.1 [32] (Table 1).

## Meta-analysis

Based on this study, the overall estimated pooled prevalence of late-stage presentation among cervical cancer patients with a random-effects model was 60.66% (95% CI: 56.27, 65.06) with a heterogeneity index ($I^2$) of 98.4% (p = 0.000) (Fig 2).

To adjust, the reported heterogeneity of the study ($I^2$ = 98.4%), the subgroup analysis based on the world region had done; accordingly, the prevalence of late-stage presentation among cervical cancer patients was found 62.60% in Africa, 69.30% in Asia, 46.51% in Europe, and 50.16% in North America (Fig 3).

Meta-regression was conducted to identify the source of heterogeneity by using sample size and year of publication as a covariate; the result showed there is no effect of sample size and year of publication on heterogeneity between studies (Table 2). A publication bias was assessed using funnel plot and objectively by Egger test at a 5% significant level. A funnel plot showed asymmetrical distribution evidencing publication bias, and the Egger tests were not statistically significant with a p-value = 0.623 (Fig 4).

**Table 1. Characteristics of the included studies in the systematic review and meta-analysis.**

| Authors Name | Publication Year | Study area | Study design | Sample size | Prevalence (95% CI) |
|---|---|---|---|---|---|
| Wassie M, Fentie B. | 2021 | Ethiopia | Cross-sectional | 1057 | 56.8(53.81–59.78) |
| Mlange R, et al. | 2016 | Tanzania | Cross-sectional | 202 | 63.9(57.27–70.52) |
| Dunyo P | 2018 | Ethiopia | Cross-sectional | 157 | 65.97(58.55–73.38) |
| Ndlovu N | 2003 | Zimbabwe | Cross-sectional | 108 | 80(72.45–87.54) |
| Ibrahim A, et al. | 2011 | Sudan | Cohort | 197 | 72(65.72–78.27) |
| Gyenwali D, et al. | 2013 | Nepal | Cross-sectional | 110 | 80.9(73.55–88.24) |
| Behnamfar F, Azadehrah M | 2015 | Iran | Cross-sectional | 55 | 89.1(80.86–97.33) |
| Dereje N, et al. | 2020 | Ethiopia | Cross-sectional | 50 | 60.4(46.84–73.95) |
| Zeleke S, et al. | 2021 | Ethiopia | Cross-sectional | 410 | 86.3(82.97–89.62) |
| Begoihn M, et al. | 2019 | Ethiopia | Cohort | 1575 | 55.2(52.74–57.65) |
| Tanturovski D, et al. | 2013 | Macedonia | Cross-sectional | 107 | 72(63.49–80.50) |
| Ouasmani F et al. | 2016 | Morocco | Cross-sectional | 401 | 39.9(35.10–44.69) |
| Panda J, et al. | 2020 | India | Cross-sectional | 122 | 39.3(30.63–47.96) |
| Berraho M, et al. | 2012 | Morocco | Cross-sectional | 200 | 54.5(47.59–61.40) |
| Frida KM, et al. | 2017 | Kenya | Cross-sectional | 152 | 53.9(45.97–61.82) |
| Kaku M, et al. | 2008 | India | Cross-sectional | 473 | 50.4(45.89–54.90) |
| Lim AW, et al. | 2014 | England | Cross-sectional | 128 | 28(20.22–35.77) |
| Ferrante JM, et al. | 2000 | USA | Cross-sectional | 852 | 45.3(41.95–48.64) |
| Friebel-Klingner TM, et al. | 2021 | Botswana | Cross-sectional | 984 | 44.7(41.59–47.80) |
| Mwaka AD, et al. | 2016 | Uganda | Cross-sectional | 149 | 65(57.34–72.65) |
| El Ibrahimi S, Pinheiro PS. | 2017 | USA | Cross-sectional | 31425 | 59(58.45–59.54) |
| Senapati R, | 2016 | India | Cross-sectional | 246 | 78.04(72.86–83.21) |
| Tiwari V, et al. | 2015 | India | Cross-sectional | 300 | 77.87(73.17–82.56) |
| Nassali MN, et al. | 2018 | Botswana | Cross-sectional | 149 | 55.1(47.11–63.08) |
| Saghari S, et al. | 2015 | USA | Cross-sectional | 13624 | 46(45.16–46.83) |

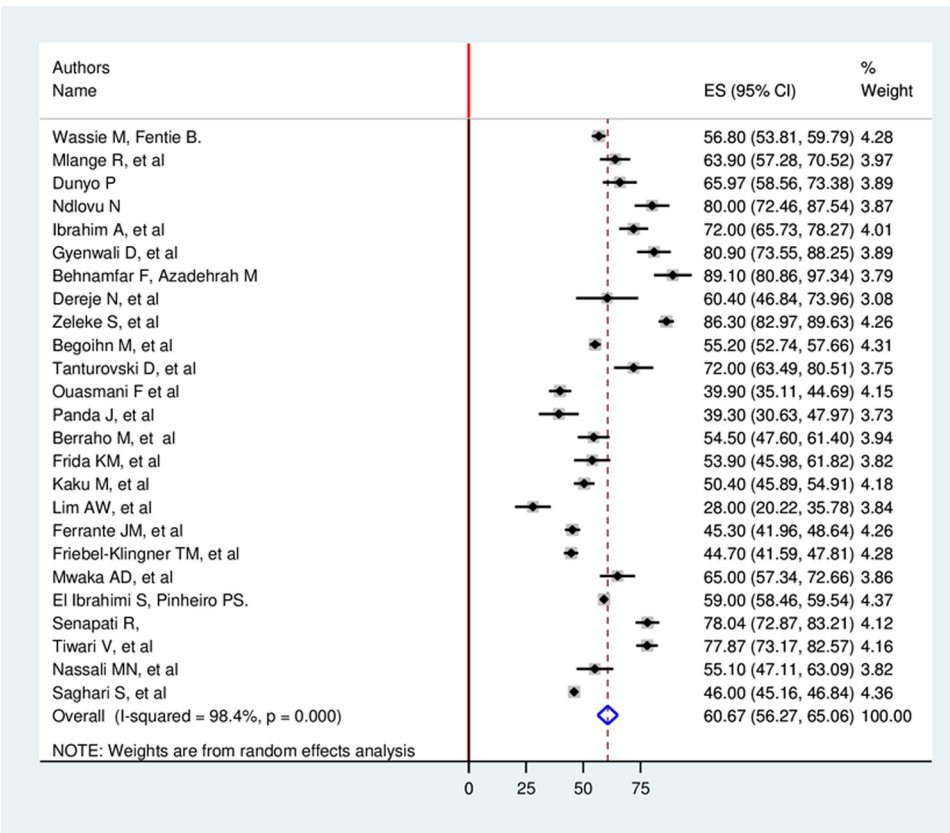

**Fig 2. Forest plot showing pooled global prevalence of late stage presentation among cervical cancer patient.**

Sensitivity analysis had carried out by removing studies step by step to evaluate the effect of a single study on the overall effect estimate but, removing of a single study wouldn't have a significant influence on pooled prevalence. Additionally, there is no study away from the lower and upper limit of confidence interval (Fig 5).

## Determinants of late stage presentation

Five variables were extracted to identify determinants of late-stage presentation among cervical cancer patients. Of those, two variables (educational status and place of residence) had identified as significant factors (Table 3).

Accordingly, those patients educated primary and above were 61% less likely to have the late presentation of cervical cancer than those with no formal education (OR: 0.39(95%CI 0.17–0.19), p = 0.031, $I^2$: 85.4%, the heterogeneity test (p< 0.001).

Those patient who came from the rural area were 2.87 times more likely to have a late presentation of cervical cancer than who come from urban areas (OR: 2.87(95%CI 1.38–5.93), p = 0.004, $I^2$: 94.9%, the heterogeneity test (p< 0.001).

## Discussion

A study showed the fate of patients after proved diagnosis of cervical cancer was 98% of the patients' consent to further medical care, and around 27% finally had a hysterectomy [51]. The figure is one indication of the worse effect of cervical cancer.

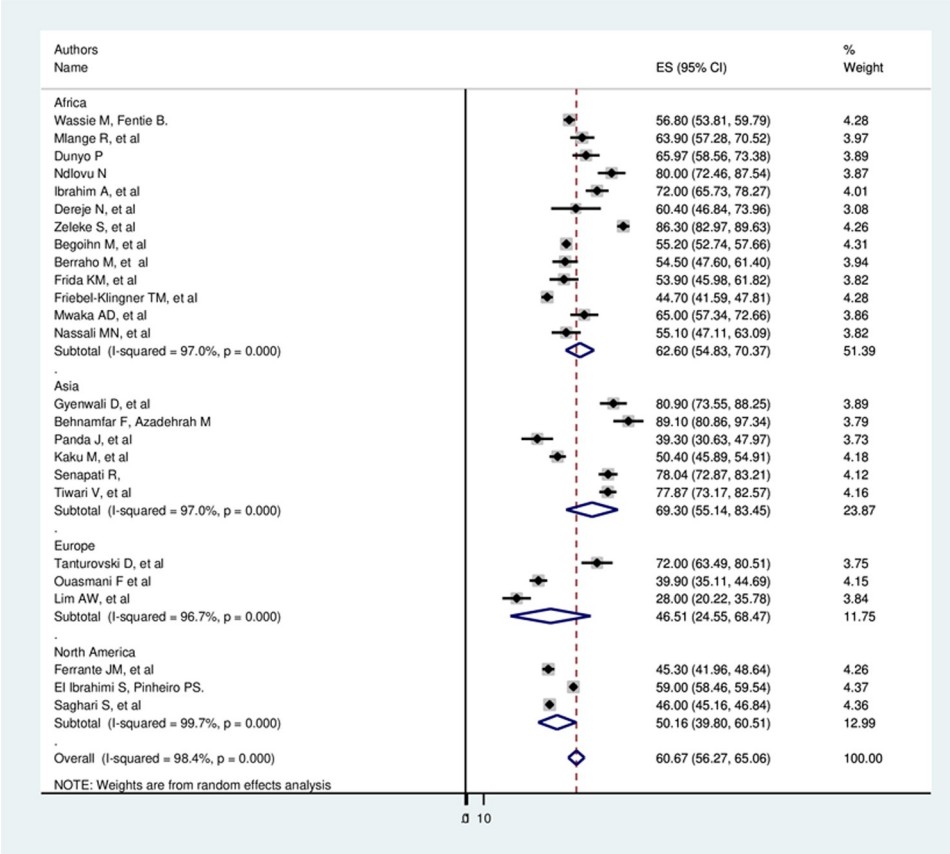

**Fig 3. Subgroup analysis of global late stage presentation among cervical cancer patients.**

According to this Meta-analysis, the overall estimated pooled prevalence of late-stage presentation among cervical cancer patients with a random-effects model was 60.66% (95% CI: 56.27, 65.06) with a heterogeneity index ($I^2$) of 98.4% (p = 0.000). The increased prevalence of late stage presentation of this preventable and curable cancer puts a patient at higher risk of death [52]. The subgroup analysis showed that the prevalence of late-stage presentation among cervical cancer patients was 62.60% in Africa, 69.30 in Asia, 46.51% in Europe, and 50.16% in North America. The highest prevalence in low-medium income countries might be due to poor urbanization level, demographic, and low socio-economic characteristics of the study participants. This systematic review and meta-analysis also identified pooled determinants of late presentation among cervical cancer patients. Among the extracted factors' educational status and place of residence are significantly associated.

There is an association between educational level and late-stage presentation. This study t also indicated, patients educated primary and above were 61% less likely to have the late stage presentation of cervical cancer than those with no formal education (OR: 0.39(95%CI 0.17–

**Table 2. Meta-regression analysis of factors affecting between-study heterogeneity on prevalence of late-stage presentation.**

| Heterogeneity source | Coefficients | Std. Err. | P-value |
|---|---|---|---|
| Sample size | 0.0000892 | .0017334 | 0.959 |
| Year of publication | 0.0231892 | 2.083469 | 0.991 |

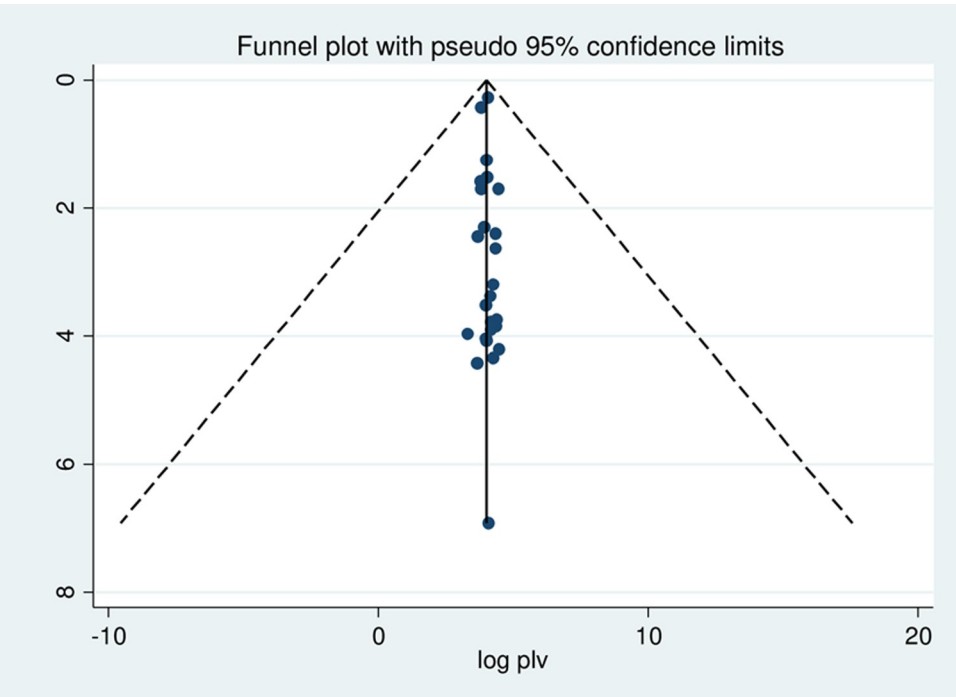

**Fig 4. Funnel plot to test the publication bias in 25 studies with 95% confidence limits.**

0.19). Different studies showed that low educational level is a risk factor and determinant of health-seeking behavior among patients [53, 54]. Those patient who came from the rural area were 2.87 times more likely to have a late -stage presentation of cervical cancer than who come from urban areas (OR: 2.87(95%CI 1.38–5.93). Place of residence determines health care

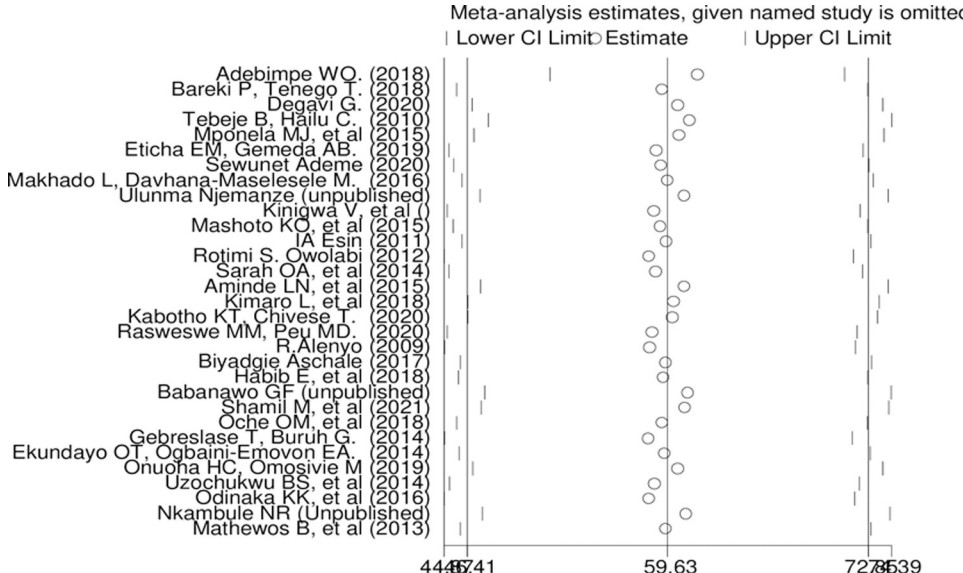

**Fig 5. Sensitivity analysis of pooled prevalence of late stage presentation for each study being removed one at a time.**

**Table 3. Determinants of late-stage presentation among cervical cancer patients.**

| Determinants | Comparison | No of studies | Sample size | OR(95%CI) | P-value | $I^2$ (%) | Heterogeneity test (p value) | Egger's test (P-value) |
|---|---|---|---|---|---|---|---|---|
| Educational status | No formal education Vs primary and above | 7 | 1014 | 0.39(0.17–0.91) | 0.031 | 85.4 | < 0.001 | 0.413 |
| Age of the patient | < 50 years old Vs > = 50 years old | 3 | 14833 | 1.91(0.94–3.90) | 0.073 | 96.0 | < 0.001 | 0.681 |
| Marital status | Married Vs others | 6 | 15735 | 1.433(0.86–2.37) | 0.162 | 92.3 | < 0.001 | 0.760 |
| Place of residence | Urban Vs Rural | 5 | 4814 | 2.87(1.38–5.93) | 0.004 | 94.9 | < 0.001 | 0.200 |
| Screening history | Yes vs No | 4 | 1194 | 2.31(0.91–5.85) | 0.077 | 72.2 | 0.013 | 0.374 |

service utilization, and in advance, it's related to premature mortality [55, 56] secondary to delayed presentation and poor health care seeking behavior.

## Conclusion

The results of this meta-analysis indicated that the prevalence of late-stage presentation of cervical cancer is substantially high. Place of residence and educational status were significantly associated with late-stage presentation. Health care organizations should work on early screening, management, and on increasing community awareness to minimize late stage at presentation among those patients.

## Limitation of the study

This systematic review and meta-analysis presented up-to-date evidence on the prevalence of late-stage presentation and determinants of cervical cancer; it might have faced the following limitations. First, the lack of studies from three regions of the world may affect the generalizability of the finding. Secondly, we have faced difficulties comparing our findings due to the lack of regional and worldwide systematic reviews and meta-analyses.

## Supporting information

**S1 Checklist. PRISMA checklist.**
(DOCX)

**S1 File. Search strategy.**
(DOCX)

**S1 Data. Data of meta-analysis.**
(XLSX)

## Acknowledgments

We would like to thank all authors of studies included in this systematic review and meta-analysis.

## Author Contributions

**Conceptualization:** Tiwabwork Tekalign.

**Data curation:** Tiwabwork Tekalign, Mister Teshome.

**Formal analysis:** Tiwabwork Tekalign, Mister Teshome.

**Investigation:** Tiwabwork Tekalign.

**Methodology:** Tiwabwork Tekalign, Mister Teshome.

**Software:** Tiwabwork Tekalign, Mister Teshome.

**Validation:** Tiwabwork Tekalign, Mister Teshome.

**Visualization:** Tiwabwork Tekalign, Mister Teshome.

**Writing – original draft:** Tiwabwork Tekalign.

**Writing – review & editing:** Tiwabwork Tekalign, Mister Teshome.

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
