## [Decision Letter · Decision Letter 0]

9 Feb 2022

PONE-D-21-23859Determinants of late-stage presentation among cervical cancer patients, a systematic review and Meta-analysisPLOS ONE

Dear Dr. Tiwabwork Tekalign, 

Thank you for submitting your manuscript to PLOS ONE. After careful consideration, we feel that it has merit but does not fully meet PLOS ONE’s publication criteria as it currently stands. Therefore, we invite you to submit a revised version of the manuscript that addresses the points raised during the review process.

We look forward to receiving your revised manuscript.

Kind regards,

Obinna Ikechukwu Ekwunife, PhD

Academic Editor

PLOS ONE

Journal Requirements:

2. Please provide the full electronic search strategy for at least one database, including any limits used, such that it could be repeated.

N/A

5. We noticed you have some minor occurrence of overlapping text with the following previous publication(s), which needs to be addressed:

- https://www.hindawi.com/journals/ogi/2016/4840762/

- http://koreascience.or.kr/journal/view.jsp?kj=POCPA9&py=2015&vnc=v16n2&sp=635

- https://www.dovepress.com/front_end/factors-associated-with-delayed-diagnosis-of-cervical-cancer-in-tikur--peer-reviewed-fulltext-article-CMAR

In your revision ensure you cite all your sources (including your own works), and quote or rephrase any duplicated text outside the methods section. Further consideration is dependent on these concerns being addressed.

Reviewers' comments:

Reviewer's Responses to Questions

**Comments to the Author**

1. Is the manuscript technically sound, and do the data support the conclusions?

Reviewer #1: Partly

Reviewer #2: No

2. Has the statistical analysis been performed appropriately and rigorously? 

Reviewer #1: Yes

Reviewer #2: No

3. Have the authors made all data underlying the findings in their manuscript fully available?

Reviewer #1: No

Reviewer #2: Yes

4. Is the manuscript presented in an intelligible fashion and written in standard English?

Reviewer #1: Yes

Reviewer #2: No

5. Review Comments to the Author

Reviewer #1: 1.The topic is not novel enough, there is no important clinical value.

2.The meta-analysis unreported preliminary design analysis.

3.Literature Searches and Search terms are incomplete. To ensure a high recall rate, otherwise it will affect the reliability and authenticity of the conclusions of Meta analysis. Please attach search terms that were used in each database as supplement for Data source and search strategies in the manuscript. Please provide details search terms in supplementary documents. Please attach syntax used in each database as supplementary.

4.There is still a considerable heterogeneity as in your limitation. Researchers can perform a subgroup analysis or Meta-regression analysis.

Reviewer #2: Thank you for the opportunity to review this paper. This is an interesting topic but there so many issues that need to be addressed. First, the language is very poor. The authors need a support from a native English language writer or seek the journal’s language editing service. Second, the study objectives and methodology were poorly written. They stated under “outcome measure” that the primary measure was “prevalence” while the secondary measure was “determinants” but the title of this paper was “determinants”. So many missing information and not adhering to PRISMA guideline. Also not registered in PROSPERO. Results and tables were poorly presented. The authors should seek support from a librarian or an expert in systematic review to review this interesting paper. Below are a few comments. I could not present all errors I found in the paper due to its fundamental flaws.

Abstract:

1. The authors need to specify that they also want to estimate the prevalence and not only the determinants.

2. Several descriptive and grammatical errors. Please have a review of the choice of word. Example first sentence in the results

“From 64,059 obtained studies, 25 studies from 3 world regions involving 53,233 participants enrolled in this meta-analysis.”

Also in the conclusion:

“…..Therefore health care organizations should work on early screening and treatment, as well as increasing community awareness to minimize premature death among those patients is essential” needs to be rephrased. The use of the term “therefore” doesn’t fit in.

Also see the Introduction section:

“Several studies worldwide have investigated the factors associated with delayed diagnosis of the cancer and disparities in its mortality rate in different racial, geographic and socio-economic groups [10-13]. However knowledge of delays for this cancer could be useful in establishing comprehensive preventative strategies.” The use of However is a problem here.

Please check other through out this paper. I found so many disconnect and wrong use of coordinating and subordinating conjunctions.

Introduction

1. I would move this statement forward to the first or second paragraph:

“In Africa, high incidences of cervical cancer are reported at rates exceeding 50 per 100,000 populations [17]. In sub-Saharan Africa, cervical cancer is the second commonest cancer morbidity and the leading cause of mortality with over 577,000 deaths annually; furthermore, in Eastern Africa it is the number one commonest cancer in women [18].”

2. “Other factors were scattered studied so, this systematic

review and meta-analysis aimed is to identify pooled determinants of late-stage presentation among cervical cancer patients”???.

This sentence is not clear.

3. Authors need to align their study objectives to the study title.

Methods

1. The methodology of this review was poorly written. The authors claim to follow the PRISMA guideline, but I could not verify this in the methodology.

2. Looks like this systematic review was not registered in PROSPERO

3. “…The databases used were EMBASE, MEDLINE, Cochrane, Scopus, Web of Science,

CINAHL, and manually on Google Scholar.” This should be under information sources.

4. “Both funnel plot and Egger’s test had used to assess publication bias. A p-value< 0.05 had used to declare the statistical significance of publication bias. Also, I2 test statistics had used to check the heterogeneity of studies. I2 test statistics of < 50, 50–75% and > 75% was declared as low, moderate and high heterogeneity respectively [25]”.

So many errors here and there and poorly written. The authors could not clearly describe their analytical approach and the reasons for what they did.

6. PLOS authors have the option to publish the peer review history of their article (what does this mean?). If published, this will include your full peer review and any attached files.

Reviewer #1: No

Reviewer #2: **Yes: **Charles Okafor

---

## [Author Response · Author response to Decision Letter 0]

3 Mar 2022

REVIEWER COMMENTS

Determinants of late-stage presentation among cervical cancer patients, a systematic review and Meta-analysis Thank you for the opportunity to review this paper. This is an interesting topic but there so many issues that need to be addressed. First, the language is very poor. The authors need a support from a native English language writer or seek the journal’s language editing service. Second, the study objectives and methodology were poorly written. They stated under “outcome measure” that the primary measure was “prevalence” while the secondary measure was “determinants” but the title of this paper was “determinants”. So many missing information and not adhering to PRISMA guideline. Also not registered in PROSPERO. Results and tables were poorly presented. The authors should seek support from a librarian or an expert in systematic review to review this interesting paper. Below are a few comments. I could not present all errors I found in the paper due to its fundamental flaws. Abstract: The authors need to specify that they also want to estimate the prevalence and not only the determinants. Response – corrected Several descriptive and grammatical errors. Please have a review of the choice of word. Example first sentence in the results “From 64,059 obtained studies, 25 studies from 3 world regions involving 53,233 participants enrolled in this meta-analysis.”Also in the conclusion: “…..Therefore health care organizations should work on early screening and treatment, as well as increasing community awareness to minimize premature death among those patients is essential” needs to be rephrased. The use of the term “therefore” doesn’t fit in.Also see the Introduction section: “Several studies worldwide have investigated the factors associated with delayed diagnosis of the cancer and disparities in its mortality rate in different racial, geographic and socio-economic groups [10-13]. However knowledge of delays for this cancer could be useful in establishing comprehensive preventative strategies.” The use of However is a problem here. Please check other through out this paper. I found so many disconnect and wrong use of coordinating and subordinating conjunctions. Response – corrected IntroductionI would move this statement forward to the first or second paragraph: “In Africa, high incidences of cervical cancer are reported at rates exceeding 50 per 100,000 populations [17]. In sub-Saharan Africa, cervical cancer is the second commonest cancer morbidity and the leading cause of mortality with over 577,000 deaths annually; furthermore, in Eastern Africa it is the number one commonest cancer in women [18].” Response – corrected “Other factors were scattered studied so, this systematicreview and meta-analysis aimed is to identify pooled determinants of late-stage presentation among cervical cancer patients”???.This sentence is not clear. Response – corrected Authors need to align their study objectives to the study title. Response – corrected MethodsThe methodology of this review was poorly written. The authors claim to follow the PRISMA guideline, but I could not verify this in the methodology.Response – corrected and followed updated PRISMA guideline Looks like this systematic review was not registered in PROSPEROResponse – corrected and registration ID was mentioned“…The databases used were EMBASE, MEDLINE, Cochrane, Scopus, Web of Science,CINAHL, and manually on Google Scholar.” This should be under information sources.Response – corrected “Both funnel plot and Egger’s test had used to assess publication bias. A p-value< 0.05 had used to declare the statistical significance of publication bias. Also, I2 test statistics had used to check the heterogeneity of studies. I2 test statistics of < 50, 50–75% and > 75% was declared as low, moderate and high heterogeneity respectively [25]”. So many errors here and there and poorly written. The authors could not clearly describe their analytical approach and the reasons for what they did. Response – corrected

---

## [Decision Letter · Decision Letter 1]

12 Apr 2022

Prevalence and determinants of late-stage presentation among cervical cancer patients, a systematic review and Meta-analysis

PONE-D-21-23859R1

Dear Dr. Tiwabwork Tekalign,

We’re pleased to inform you that your manuscript has been judged scientifically suitable for publication and will be formally accepted for publication once it meets all outstanding technical requirements.

Kind regards,

Obinna Ikechukwu Ekwunife, PhD

Academic Editor

PLOS ONE

Additional Editor Comments (optional):

Reviewers' comments:

Reviewer's Responses to Questions

**Comments to the Author**

1. If the authors have adequately addressed your comments raised in a previous round of review and you feel that this manuscript is now acceptable for publication, you may indicate that here to bypass the “Comments to the Author” section, enter your conflict of interest statement in the “Confidential to Editor” section, and submit your "Accept" recommendation.

Reviewer #1: All comments have been addressed

Reviewer #2: All comments have been addressed

2. Is the manuscript technically sound, and do the data support the conclusions?

Reviewer #1: Partly

Reviewer #2: Yes

3. Has the statistical analysis been performed appropriately and rigorously? 

Reviewer #1: I Don't Know

Reviewer #2: Yes

4. Have the authors made all data underlying the findings in their manuscript fully available?

Reviewer #1: Yes

Reviewer #2: Yes

5. Is the manuscript presented in an intelligible fashion and written in standard English?

Reviewer #1: Yes

Reviewer #2: Yes

6. Review Comments to the Author

Reviewer #1: (No Response)

Reviewer #2: My concerns have been addressed. I have no further comments. The figures may need editing for qauality.

7. PLOS authors have the option to publish the peer review history of their article (what does this mean?). If published, this will include your full peer review and any attached files.

Reviewer #1: No

Reviewer #2: No

---

## [Editor Report · Acceptance letter]

18 Apr 2022

PONE-D-21-23859R1 

Prevalence and determinants of late-stage presentation among cervical cancer patients, a systematic review and Meta-analysis 

Dear Dr. Tekalign:

I'm pleased to inform you that your manuscript has been deemed suitable for publication in PLOS ONE. Congratulations! Your manuscript is now with our production department. 

Kind regards, 

on behalf of

Dr. Obinna Ikechukwu Ekwunife 

Academic Editor

PLOS ONE